# Credit Risk Evaluation of Forest Farmers under Internet Crowdfunding Mode: The Case of China's Collective Forest Regions

Xiaowo Wu [1], Jiangwei Tu [2], Boru Liu [2], Xi Zhou [2,*] and Yanxiong Wu [2,*]

[1] College of Sciences and Engineering, University of Tasmania, Hobart, TAS 7001, Australia; wu20086@gmail.com
[2] Jiyang College, Zhejiang A&F University, Zhuji 311800, China; tujiangweisax@sina.com (J.T.); liuboru_2022@163.com (B.L.)
[*] Correspondence: zhouxi654321@126.com (X.Z.); 13601213097@163.com (Y.W.)

**Abstract:** To effectively quantify and control the credit risk of forest farmers under internet crowdfunding mode, the combined weighting of norm grey correlation, the improved analytic hierarchy process and empirical mode decomposition method are proposed to measure the credit risk, and the interval rough number DEMATEL method is used to analyze the credit risk factors of forest farmers. Through the calculation of comprehensive influence degree, it is concluded that the degree of investor information asymmetry, the intensity of supervision, the degree of innovation and cooperation between funders and investors are the main credit risk factors of forest farmers under internet crowdfunding mode, and a credit risk control mechanism is constructed according to the main credit risk factors to effectively improve the risk management and control level of forest farmers.

**Keywords:** internet crowdfunding; forest farmers; credit risk evaluation; combination weighting; immunity





## 1. Introduction

By the end of 2013, the reform of the main body of the national collective forest right system with clear property rights and household contract responsibility had been basically completed. According to statistics, the ownership of nearly 180 million hm$^2$ of collective forest land has been confirmed by households, accounting for 99% of the total area of the collective forest land, and about 90 million forest farmers have obtained the forest property rights certificate [1]. Through the reform, forest farmers have obtained the usufruct of forest land, the ownership of forest trees, and the autonomy and income right of forest land management, and thus their production enthusiasm has been significantly improved. However, financing difficulty is still the key problem restricting the sustainable management of forests. Online crowdfunding, a financing mode of internet finance, provides a new option to solve the financing difficulties of forest farmers.

According to the research of Xiaoqing Ma et al. and Y. Wang et al., the credit risks of forest farmers mainly include liquidity risk, credit risk and operational risk, of which credit risk is the most important credit loan factor [2,3]. Q. Tang et al. and Jian Zhu et al. indicate that because the credit of forest farmers is mainly affected by farmers themselves, banks and external factors, it is often difficult to effectively meet the credit needs of forest farmers [4,5]. Ting Sun et al., by econometric methods [6], conduct a regression analysis on the factor of forestry characteristics affecting the credit constraints of forest farmers; using the game model method, Youliang Ning et al., Wenmei Liao et al. and Guanjun Huang et al. make a theoretical analysis of the game behavior between forest farmers and banks in the process of mortgage loan [7–9]. Haohui Xiao and Junxiang Li et al. evaluate the credit risk of forest farmers based on the concepts and the decision-making methods of grey system correlation and grey target [10,11]. Ling Zhu et al. and Yuxin Zhou et al. conclude that

the commonly used comprehensive benefit evaluation methods include grey correlation analysis, analytic hierarchy process, AHP (Analytic Hierarchy Process)-fuzzy evaluation, principal component analysis, etc. [12,13].

With different emphases in the analysis of forest farmer' credit risk evaluation, most of the literature research focuses on the game model, the correlation model of the grey system, econometrics and so on. At present, using the objective weighting method only in credit risk evaluation may lose the rationality of the evaluation results due to its mechanical dependence on data. What is more, objective data is difficult to obtain, so the data collected or mined is often small in amount and incomplete [14–17]. On the other hand, using the subjective weighting method only may cause problems, such as uncertainty due to its over reliance on personal experience; therefore, it is difficult to achieve the effect of accurate evaluation by using only one method [18–21]. This study uses a combined subjective-objective weighting method, which features good interpretability, strong stability and process visibility. This evaluation method not only combines the opinions of experts, but also highlights at the same time both the advantages of the subjective weighting method in terms of considering and expressing experts' subjective real intention and experience [22,23], and the advantages of the objective weighting method in terms of objectivity and high precision of credit risk evaluation. Using a combination weighting method and interval rough number DEMATEL method, this paper analyzes the credit risk of forest farmers under internet crowdfunding mode and constructs a set of scientific credit risk evaluation models based on the combined weighting method [24–29].

## 2. Materials and Methods

### 2.1. Credit Risk Evaluation Method Based on Norm Grey Correlation

In this study, the credit data of 2132 forest farmers are obtained from the online loan platform of third-party credit investigation by means of GooSeeker web crawler and other technical means, and among them, 117 forest farmers have had default records of online crowdfunding (overdue records); with special focus on the influencing factors of credit risk among forest farmers, and considering such aspects as the regions of these farmers, the industry characteristics and industry balance, this study selects 36 representative forest farmers with online crowdfunding default records for investigation (17 secondary indicators are designed based on the literature research [30–45]). Based on the survey results, 30 experts in relevant fields are interviewed. According to the analysis of the operation mode of online crowdfunding, combined with the financial data of forest farmers in the Wande information database, 11 secondary indicators, as shown in Table 1, are selected from the 17 secondary indicators, and are classified into three primary indicators: investor [30–34], fundraiser (forest farmers) [35–40] and crowdfunding platform [41–44]. Among the secondary indicators, the investor's cognitive ability refers to the investor's ability to grasp and understand forest farmers' extraction and storage of information, direction and driving force of business development, so as to make more rational decisions. Other indicators are easy to understand and will not be explained separately. The specific indicators are shown in Table 1:

Ten typical forest farmers who made credit under internet crowdfunding mode are selected. According to the primary and secondary indicator systems of forest farmers' credit risk under internet crowdfunding mode, the data of the 10 forest farmers are collected and analyzed by the norm grey correlation method. The specific calculation is as follows:

- Consistency processing

For the intermediate indicator of forest farmers' relevant indicators

$$b_{x'} = \begin{cases} b_x - m & j_m \leq x \leq j_a \\ b_x & b_x = j_a \\ b_M - b_x & j_a < b_x \leq b_M \end{cases} \tag{1}$$

**Table 1.** Forest farmers' credit risk evaluation indicator system under internet crowdfunding mode.

| Target Layer | Primary Indicators | Secondary Indicators |
|---|---|---|
| Credit Risk Measurement Indicator System ($D$) | Investor ($D_1$) | Investor's cognitive ability ($D_{11}$) <br> Investor's supervision ability ($D_{12}$) |
| | Fundraiser ($D_2$) | Financing duration and scope ($D_{21}$) <br> Degree of innovation and cooperation with the investor ($D_{22}$) <br> Degree of information asymmetry with the crowdfunding platform ($D_{23}$) <br> Degree of information asymmetry with the investor ($D_{24}$) <br> Withdrawal difficulty ($D_{25}$) |
| | Crowdfunding platform ($D_3$) | Margin ratio ($D_{31}$) <br> Service fee proportion ($D_{32}$) <br> Intensity of regulation ($D_{33}$) <br> Number of media reports ($D_{34}$) |

In the formula: $b_M$ is the upper bound of the indicator $b_x$, $j_m$ is the lower bound of the indicator $h_x$, and $j_a$ is a fixed value.

For the interval indicator of forest farmers' relevant indicators

$$b_{x'} = \begin{cases} 1 - \frac{zq_1 - b_x}{\max\{zq_1 - j_m, b_M - zq_2\}} & b_x < zq_1 \\ 1 & zq_1 \leq b_x \leq zq_2 \\ 1 - \frac{zq_1 - b_x}{\max\{zq_1 - j_m, b_M - zq_2\}} & b_x > zq_2 \end{cases} \tag{2}$$

$zq_1 \leq b_x \leq zq_2$ is the most stable interval of the indicator $b_x$, $b_M$ is the upper bound of the indicator $b_x$, and $j_m$ is the lower bound of the indicator $b_x$.

- Dimensionless processing

$$w_{z_{ij}} = \frac{b_{x_{is_j}}}{b_{x_{0s_j}}} \tag{3}$$

$b_{x_{0s_j}}$ is the reference value of the indicator $s_j$. If there is no definite reference value for the indicator, then

$$b_{x_{0s_j}} = \max_i(b_{x_{s_j}}) \tag{4}$$

- Determining the indicator weight

For indicator $s_j$, by taking the indicator value sequence of the other indicator $s_k$ as the reference sequence and the indicator sequence of indicator $s_j$ as the comparison sequence, the norm grey correlation degree $\rho_{s_j s_k}(1 \leq s_j \leq n, 1 \leq s_k \leq n)$ of indicator $s_j$ relative to the other indicator $s_k$ is obtained, in which $\rho_{s_j s_k} = 1$, that is, the indicator has the greatest correlation with itself.

Let $r_i(s_j, s_k)$ be the $i$ th grey correlation coefficient of indicator $s_j$ to indicator $s_k$

$$r_i(s_j, s_k) = \frac{\min\limits_{s_j} \min\limits_{g \in i} \left| y_{s_k}(g) - y_{s_j}(g) \right| + \zeta \max\limits_{s_j} \max\limits_{g \in i} \left| y_{s_k}(g) - y_{s_j}(g) \right|}{y_{s_k}(g) - y s_j(g) + \zeta \max\limits_{s_j} \max\limits_{g \in i} \left| y_{s_k}(g) - y_{s_j}(g) \right|} \tag{5}$$

In the formula, $y_{s_k}(g)$ and $y_{s_j}(g)$ represent the $g$ th indicator value corresponding to indicator $s_k$ and indicator $s_j$, respectively, and $\zeta$ is 0.5 under the principle of minimum information. Let $\rho = r_i(s_j, s_k)$

Positive ideal sequence of correlation coefficient:

$$\rho^+ = \left\{ \max b_{x_{s_j}} \middle| 1 \leq s_j \leq n, 1 \leq s_k \leq n \right\} = \left\{ \rho^+(s_k) \middle|, 1 \leq s_k \leq n \right\} \tag{6}$$

Negative ideal sequence of correlation coefficient:

$$\rho^- = \left\{ \min b_{x_{s_j}} \middle| 1 \leq s_j \leq n, 1 \leq s_k \leq n \right\} = \left\{ \rho^-(s_k) \middle|, 1 \leq s_k \leq n \right\} \tag{7}$$

The two norms of the norm grey correlation's determinant of coefficient $(\rho_{s_{j1}}, \rho_{s_{j2}}, \cdots, \rho_{s_{jn}})$ of the $s_j$ th indicator sequence are defined as:

$$d_{s_j}^+ = \sqrt{\sum_{s_k=1}^{n} \left( \rho_{s_j n} - \rho^+(s_k) \right)^2} \tag{8}$$

$$d_{s_j}^- = \sqrt{\sum_{s_k=1}^{n} \left( \rho_{s_j n} - \rho^-(s_k) \right)^2} \tag{9}$$

The group norm grey correlation degree of the related indicator $s_j$ of forest farmers is

$$\rho_{s_j} = \frac{d_{s_j}^-}{d_{s_j}^+ + d_{s_j}^-} \tag{10}$$

The weight coefficient of the related indicators of forest farmers is

$$z\omega_{s_j} = \frac{\rho_{s_j}}{\sum\limits_{s_j=1}^{n} \rho_{s_j}} \tag{11}$$

The final weight coefficients of each indicator according to the calculation of Equations (1)–(11) are shown in Table 2.

**Table 2.** Weight coefficient of credit risk evaluation indicator system.

| $D_{11}$ | $D_{12}$ | $D_{21}$ | $D_{22}$ | $D_{23}$ | $D_{24}$ | $D_{25}$ | $D_{31}$ | $D_{32}$ | $D_{33}$ | $D_{34}$ |
|----------|----------|----------|----------|----------|----------|----------|----------|----------|----------|----------|
| 0.0518 | 0.0424 | 0.0919 | 0.0943 | 0.0665 | 0.1924 | 0.0426 | 0.0718 | 0.0927 | 0.1822 | 0.0714 |

*2.2. Credit Risk Evaluation Method Based on Improved Analytic Hierarchy Process and Empirical Mode Decomposition*

The traditional analytic hierarchy process uses a 1–9 scale method to establish a judgment matrix and thus has the following main shortcomings: there is a large gap between the ranking result and people's psychological judgment; the consistency of the judgment matrix conflicts with the consistency of thinking. The consistency matrix has poor construction ability, which may be in reverse order with the actual ranking; the scale value is inconsistent with the ranking method; the mathematical structure is poor in nature, and so on. Considering the nonstationarity and nonlinearity of credit risk indicators, the difficulty in vividly describing the expert judgment process, as well as the fuzziness of the relationship between credit risk influencing factors, combined with the triangular fuzzy number and semantic transformation, this study tries to standardize and clarify the credit risk indicators of online crowdfunding to obtain the comprehensive evaluation decision matrix, which integrates the decision-making thinking of different experts to avoid the subjectivity of evaluation and further reveals the most critical factors affecting credit risk.

Furthermore, in the evaluation of the credit risk indicators of forest farmers under network crowdfunding, the improved analytic hierarchy process and empirical mode

decomposition method are used in this study to determine the weight of the indicators in hope of making the evaluation more scientifically. The steps we take are as follows:

- Construct triangular fuzzy number scale

Ten experts in relevant industries are numbered as $\left\{ z^{h_1}, z^{h_2}, \cdots, z^{h_{10}} \right\}$, who adopt triangular fuzzy numbers to make the evaluation, as shown in Table 3:

**Table 3.** Comparison table of triangular fuzzy number scale.

| Comparison of the Significance of A and B | Description | Triangular Fuzzy Number |
|---|---|---|
| Equally important | By comparison between the two elements, they are equally important | (1,1,1) |
| Almost as important | By comparison between the two elements, one is a bit more important than the other | (1/2,1,2) |
| Slightly important | By comparison between the two elements, one is more important than the other | (2,3,4) |
| Obviously important | By comparison between the two elements, one is obviously more important than the other | (4,5,6) |
| Strongly important | By comparison between the two elements, one is much more important than the other | (6,7,8) |
| Extremely important | By comparison between the two elements, one is absolutely more important than the other | (8,9,9) |
| Inverse comparison | Reciprocal representation of the degree of unimportance of the two compared elements | Corresponding inverse value |

The fuzzy judgment matrix given by $z^{h_1}$ is shown in Table 4; due to the limited space of this paper, the fuzzy judgment matrix given by other experts is omitted.

- Calculate experts' evaluation value

According to LFPP (Logarithmic Fuzzy Priority Programming) target programming model:

$$\min \varsigma = (1 - \vartheta)^2 + M_m \sum_{i=1}^{n-1} \sum_{j=i+1}^{n} (\phi_{ij}^2 + \delta_{ij}^2)$$

$$\begin{cases} \ln w_i - \ln w_j - \vartheta \ln\left(\frac{b^{m_{ij}}}{d^{l_{ij}}}\right) + \phi_{ij} \geq \ln d^{l_{ij}} \\ -\ln w_i + \ln w_j - \vartheta \ln\left(\frac{u_{ij}^g}{m_{ij}}\right) + \delta_{ij} \geq -\ln u_{ij}^g \\ \vartheta \geq 0, \ln w_j \geq 0, i = 1, \cdots, n \\ \phi_{ij} \geq 0, \Pi_{ij} \geq 0, i = 1, \cdots, n-1; j = i+1, \cdots, n \end{cases} \quad (12)$$

The triangular fuzzy number is $(d^l, u^g, b^m)$; $d^l$ and $b^m$ are the left and right endpoints of the triangular fuzzy function, respectively, where $\vartheta = \min u_{ij}^g (\ln(\frac{w_i}{w_j}))$, $\phi_{ij}$ and $\delta_{ij}$ represent the deviation variables of the above constraints, and $w_i$ is the evaluation value of experts, generally $M_a = 10^{11}$. Calculated by MATLAB, the evaluation values of expert $z^{h_1}$ are 2.5654, 2.6681, 3.8762, 3.9592, 2.7873, 4.9469, 2.2217, 2.4568, 2.8763, 4.5768, 2.8764, respectively.

The evaluation values of other experts are calculated in the same way, and the results are shown in Table 5.

**Table 4.** The fuzzy judgment matrix of expert $z^{h_1}$.

| Indicators | $D_{11}$ | $D_{12}$ | $D_{21}$ | $D_{22}$ | $D_{23}$ | $D_{24}$ | $D_{25}$ | $D_{31}$ | $D_{32}$ | $D_{33}$ | $D_{34}$ |
|---|---|---|---|---|---|---|---|---|---|---|---|
| $D_{11}$ | (1,1,1) | (6,7,8) | (2,3,4) | (1/2,1,2) | (2,3,4) | (1/9,1/9,1/8) | (2,3,4) | (8,9,9) | (2,3,4) | (1/4,1/3,1/2) | (2,3,4) |
| $D_{12}$ | (1/8,1/7,1/6) | (1,1,1) | (6,7,8) | (4,5,6) | (1/2,1,2) | (1/2,1,2) | (1/8,1/7,1/6) | (1/6,1/5,1/4) | (6,7,8) | (1/2,1,2) | (1/2,1,2) |
| $D_{21}$ | (1/4,1/3,1/2) | (1/6,1/5,1/4) | (1,1,1) | (1/4,1/3,1/2) | (8,9,9) | (2,3,4) | (1/9,1/9,1/8) | (1/6,1/5,1/4) | (6,7,8) | (1/2,1,2) | (1/9,1/9,1/8) |
| $D_{22}$ | (1/2,1,2) | (1/6,1/5,1/4) | (2,3,4) | (1,1,1) | (1/4,1/3,1/2) | (4,5,6) | (1/4,1/3,1/2) | (1/2,1,2) | (2,3,4) | (1/4,1/3,1/2) | (2,3,4) |
| $D_{23}$ | (1/4,1/3,1/2) | (1/2,1,2) | (1/9,1/9,1/8) | (2,3,4) | (1,1,1) | (2,3,4) | (1/8,1/7,1/6) | (4,5,6) | (1/8,1/7,1/6) | (1/2,1,2) | (6,7,8) |
| $D_{24}$ | (8,9,9) | (1/2,1,2) | (1/4,1/3,1/2) | (1/6,1/5,1/4) | (1/4,1/3,1/2) | (1,1,1) | (4,5,6) | (1/8,1/7,1/6) | (4,5,6) | (1/4,1/3,1/2) | (8,9,9) |
| $D_{25}$ | (1/4,1/3,1/2) | (6,7,8) | (8,9,9) | (2,3,4) | (6,7,8) | (1/6,1/5,1/4) | (1,1,1) | (8,9,9) | (2,3,4) | (1/4,1/3,1/2) | (1/2,1,2) |
| $D_{31}$ | (1/9,1/9,1/8) | (4,5,6) | (4,5,6) | (1/2,1,2) | (1/6,1/5,1/4) | (6,7,8) | (1/9,1/9,1/8) | (1,1,1) | (8,9,9) | (1/6,1/5,1/4) | (6,7,8) |
| $D_{32}$ | (1/4,1/3,1/2) | (1/8,1/7,1/6) | (1/8,1/7,1/6) | (1/4,1/3,1/2) | (6,7,8) | (1/6,1/5,1/4) | (1/4,1/3,1/2) | (1/9,1/9,1/8) | (1,1,1) | (1/6,1/5,1/4) | (6,7,8) |
| $D_{33}$ | (2,3,4) | (1/2,1,2) | (1/2,1,2) | (2,3,4) | (1/2,1,2) | (2,3,4) | (2,3,4) | (4,5,6) | (4,5,6) | (1,1,1) | (1/2,1,2) |
| $D_{34}$ | (1/4,1/3,1/2) | (1/2,1,2) | (8,9,9) | (1/4,1/3,1/2) | (1/8,1/7,1/6) | (1/9,1/9,1/8) | (1/2,1,2) | (1/8,1/7,1/6) | (1/8,1/7,1/6) | (1/2,1,2) | (1,1,1) |

**Table 5.** Experts' evaluation values.

| Sequence Number | $D_{11}$ | $D_{12}$ | $D_{21}$ | $D_{22}$ | $D_{23}$ | $D_{24}$ | $D_{25}$ | $D_{31}$ | $D_{32}$ | $D_{33}$ | $D_{34}$ |
|---|---|---|---|---|---|---|---|---|---|---|---|
| Expert $z^{h1}$ | 2.5654 | 2.6681 | 3.8762 | 3.9592 | 2.7873 | 4.9469 | 2.2217 | 2.4568 | 2.8763 | 4.5768 | 2.8764 |
| Expert $z^{h2}$ | 2.6703 | 1.9853 | 3.7527 | 3.5607 | 2.7910 | 4.5977 | 2.5586 | 2.5295 | 3.5977 | 4.4970 | 2.3698 |
| Expert $z^{h3}$ | 2.4705 | 2.5653 | 3.4505 | 3.4347 | 2.5000 | 4.7038 | 2.6638 | 2.6365 | 3.7038 | 4.4393 | 2.5653 |
| Expert $z^{h4}$ | 2.4505 | 2.5000 | 3.5000 | 3.4706 | 2.3208 | 4.4414 | 2.5000 | 2.4706 | 3.6840 | 4.3794 | 2.5000 |
| Expert $z^{h5}$ | 2.3896 | 2.5826 | 3.5294 | 3.5000 | 2.3493 | 4.4705 | 2.5294 | 2.5000 | 3.6701 | 4.4604 | 2.5826 |
| Expert $z^{h6}$ | 2.2962 | 2.4705 | 3.6792 | 3.6507 | 2.5000 | 4.6206 | 2.6792 | 2.6507 | 3.5000 | 4.3493 | 2.4705 |
| Expert $z^{h7}$ | 2.3362 | 2.5000 | 3.5586 | 3.5295 | 3.3794 | 4.5000 | 2.5586 | 2.5295 | 3.5295 | 4.3794 | 2.5000 |
| Expert $z^{h8}$ | 2.1374 | 2.5873 | 3.0180 | 3.7958 | 2.1876 | 4.1374 | 2.1569 | 2.4833 | 3.7971 | 4.6544 | 2.7899 |
| Expert $z^{h9}$ | 2.2414 | 2.4043 | 3.3453 | 3.0056 | 2.5873 | 4.2414 | 2.3586 | 2.3453 | 3.7867 | 4.4543 | 2.6000 |
| Expert $z^{h10}$ | 2.4048 | 2.7871 | 3.1374 | 3.2743 | 2.8700 | 4.4048 | 2.4833 | 2.1374 | 3.6544 | 4.7871 | 2.3386 |

Improved analytic hierarchy process and the decomposition model of the empirical mode decomposition method.

The EMD (Empirical Mode Decomposition) method is an analysis method to deal with nonlinear and non-stationary signals. Differing from the general signal processing methods, it is an adaptive analysis method. In this study, $b^m = 100$ times White Gaussian Noise is added to the information series $w_i$ to obtain 100 new sets of information sequences, which are then decomposed by EMD to obtain the mean value of IMF (Intrinsic Mode Functions) components. IMF refers to the signal components of each layer obtained after the original signal is decomposed by EMD.

After $\overline{IMF}_{i1}$, the first mean value of the components, is obtained, IMF is decomposed once again to obtain the second mean value of IMF components, and then there is:

$$\overline{IMF}_2 = \frac{1}{b^m} \sum_{i=1}^{b^m} E_1(r_1(t) + \delta_1 E_1(w(i))) \tag{13}$$

Repeat the above steps and continue to decompose until the value of the average envelope is 0 to obtain the following component value:

$$\overline{IMF}_{k+1} = \frac{1}{a^p} \sum_{i=1}^{a^p} E_k(r_k(t) + \delta_1 E_k(w(i))) \tag{14}$$

Similarly, the residual component information sequence $r_i(D_i)(i = 1, 2, 3, \cdots, 11)$ of the 11 indicators can be obtained, and the final objective trend weight is as follows:

$$\overline{w}_i = \frac{\sqrt[b^q]{\prod_{t=1}^{b^q} r_i(D_i)}}{\sum_{j=1}^{b^m} \sqrt[b^q]{\prod_{t=1}^{b^q} r_i(D_i)}} \tag{15}$$

In this, $i = 1, 2, 3, \cdots, 11$.

According to the above steps, the weight coefficient is finally obtained as shown in Table 6.

**Table 6.** Weight coefficient of the indicator system.

| $D_{11}$ | $D_{12}$ | $D_{21}$ | $D_{22}$ | $D_{23}$ | $D_{24}$ | $D_{25}$ | $D_{31}$ | $D_{32}$ | $D_{33}$ | $D_{34}$ |
|---|---|---|---|---|---|---|---|---|---|---|
| 0.0412 | 0.0429 | 0.0878 | 0.0956 | 0.0735 | 0.2011 | 0.0459 | 0.0741 | 0.0933 | 0.1724 | 0.0722 |

Considering the distinguishability and interpretability of weight and reducing subjective randomness, the combination weighting formula based on the entropy weight method, the improved analytic hierarchy process and the empirical mode decomposition method is as follows:

$$w'_{z^s} = \alpha w_{zj} + \beta w_{z^q} \tag{16}$$

For the above 10 experts, the Delphi method is used to discuss the weight coefficients of $\alpha, \beta$, then the answers of each expert are summarized and corrected, and the opinions of the expert group are added up and then fed back to the expert group for further opinions. This process goes round and round, and finally, a consensus is reached, resulting in $\alpha = 0.4, \beta = 0.6$. The combined weight coefficient of forest farmers' credit risk under internet crowdfunding is thus obtained, as shown in Table 7.

**Table 7.** Weight coefficient of combined weighting of the indicator system.

| Indicators | $D_{11}$ | $D_{12}$ | $D_{21}$ | $D_{22}$ | $D_{23}$ | $D_{24}$ | $D_{25}$ | $D_{31}$ | $D_{32}$ | $D_{33}$ | $D_{34}$ |
|---|---|---|---|---|---|---|---|---|---|---|---|
| Weight Coefficient | 0.0454 | 0.0427 | 0.0894 | 0.0951 | 0.0707 | 0.1976 | 0.0446 | 0.0732 | 0.0931 | 0.1763 | 0.0719 |

This paper creatively uses the improved analytic hierarchy process to construct a set of credit risk measurement systems with high scientificity and reliability. The consistency condition is transformed into an intuitionistic fuzzy evaluation value to construct the credit risk evaluation decision matrix, thus avoiding the one-sidedness of expert weight and ensuring the authenticity of credit risk measurement results.

### 2.3. Analysis of Credit Risk Evaluation by Interval Rough Number—DEMATEL Method

In the real credit risk evaluation, the relationship between the credit risk factors of forest farmers is complex and difficult to achieve accuracy. The traditional credit risk measurement method is limited to the real number field and is often not suitable to describe the characteristics of the credit risk of forest farmers and the complex and fuzzy relationship between the credit risk factors. Since interval numbers can be used as a tool to describe the complex relationship between credit risk factors, it has the advantages of describing complex phenomena more objectively and better handling the uncertainty of evaluation.

In order to avoid the uncertainty, complexity and incomplete information in the evaluation of credit risk, an interval rough number is used to analyze the credit risk, and the decision-making thoughts of different experts are integrated to avoid the subjectivity of evaluation. This study constructs a rough set combined with the interval number DEMATEL method to solve the evaluation problems, such as the complex influence relationship between the factors in the credit risk system and the difficulty of identifying credit risk factors under the network crowdfunding mode.

An interval rough set is a rough set in which both the upper approximation and the lower approximation are intervals, which is recorded as $[a\Gamma, b\Gamma], [c\Gamma, d\Gamma]$, where $c\Gamma \leq a\Gamma \leq d\Gamma$, $\varepsilon = ([a\Gamma, b\Gamma], [c\Gamma, d\Gamma])$, $\varepsilon_1 = ([as_j, b_1\Gamma], [c_1\Gamma, d_1\Gamma])$, and $\varepsilon_2 = ([a_2\Gamma, b_2\Gamma], [c_2\Gamma, d_2\Gamma])$ are all interval rough sets, and $\lambda > 0$ is a real number, then there is:

$$\varepsilon_1 + \varepsilon_2 = ([a_1\Gamma + a_2\Gamma, b_1\Gamma + b_2\Gamma], [c_1\Gamma + c_2\Gamma, d_1\Gamma + d_2\Gamma]) \tag{17}$$

$$\lambda\varepsilon = \left\{ \begin{array}{l} [\lambda a\Gamma, \lambda b\Gamma], [\lambda c\Gamma, \lambda d\Gamma], \lambda \geq 0 \\ [\lambda b\Gamma, \lambda a\Gamma], [\lambda d\Gamma, \lambda c\Gamma], \lambda < 0 \end{array} \right\} \tag{18}$$

Step 1: Invite experts to evaluate the correlation between 11 factors according to the forest farmers' credit risk indicator system in Table 1 and establish the expert judgment matrix under interval rough numbers. Let the expert judgment matrix be $B\Gamma$, as shown in Table 8.

**Table 8.** Experts' judgment matrix.

| Influencing Factors | $D_{11}$ | $D_{12}$ | $D_{21}$ | $D_{22}$ | $D_{23}$ | $D_{24}$ | $D_{25}$ | $D_{31}$ | $D_{32}$ | $D_{33}$ | $D_{34}$ |
|---|---|---|---|---|---|---|---|---|---|---|---|
| $D_{11}$ | (0,0) | (1,3) | (1,2) | (1,3) | (3,4) | (3,5) | (2,3) | (4,5) | (4,5) | (2,3) | (1,3) |
| $D_{12}$ | (3,4) | (0,0) | (2,3) | (3,4) | (2,4) | (2,4) | (4,5) | (2,3) | (0,0) | (3,4) | (3,5) |
| $D_{21}$ | (3,5) | (3,5) | (0,0) | (3,4) | (3,5) | (0,0) | (3,4) | (3,5) | (0,0) | (0,0) | (2,3) |
| $D_{22}$ | (1,3) | (2,4) | (3,4) | (0,0) | (4,5) | (3,5) | (2,4) | (4,5) | (3,4) | (1,3) | (4,5) |
| $D_{23}$ | (0,0) | (2,3) | (3,5) | (2,3) | (0,0) | (2,3) | (3,5) | (0,1) | (2,3) | (2,3) | (0,1) |
| $D_{24}$ | (3,5) | (0,0) | (1,3) | (3,4) | (3,5) | (0,0) | (0,1) | (2,3) | (1,3) | (1,3) | (2,3) |
| $D_{25}$ | (2,3) | (3,5) | (3,4) | (3,5) | (0,0) | (2,3) | (0,0) | (3,4) | (3,5) | (1,3) | (2,3) |
| $D_{31}$ | (3,5) | (2,3) | (3,4) | (3,5) | (2,3) | (2,3) | (0,0) | (0,0) | (2,3) | (2,3) | (0,2) |
| $D_{32}$ | (3,4) | (2,3) | (3,4) | (0,0) | (2,3) | (1,3) | (2,3) | (2,3) | (0,0) | (0,0) | (1,3) |
| $D_{33}$ | (3,5) | (0,0) | (3,5) | (1,2) | (3,5) | (2,3) | (1,3) | (3,5) | (1,2) | (0,0) | (1,2) |
| $D_{34}$ | (3,5) | (2,3) | (3,5) | (0,0) | (1,3) | (2,4) | (0,0) | (3,5) | (1,2) | (0,2) | (0,0) |

Let $\varepsilon_i = ([a_i\Gamma, b_i\Gamma], [c_i\Gamma, d_i\Gamma])(i = 1, 2, 3, \cdots, n)$, $IRWA : I^n \to I$, if:

$$IRWA_w(\varepsilon_1, \varepsilon_2, \cdots, \varepsilon_n) = \sum_{i=1}^{n} w_i \varepsilon_i = \left(\left[\sum_{i=1}^{n} w_i a_i \Gamma, \sum_{i=1}^{n} w_i b_i \Gamma\right], \left[\sum_{i=1}^{n} w_i c_i \Gamma, \sum_{i=1}^{n} w_i d_i \Gamma\right]\right) \quad (19)$$

then $IRWA_w$ is called the weighted averaging operator of interval number rough set, where $I$ is the rough set of all interval numbers, $w_i$ is the weight of $\varepsilon_i$, and $0 \le w_i \le 1$, $\sum_{i=1}^{n} w_i \Gamma = 1$.

Let $\varepsilon = ([a\Gamma, b\Gamma], [c\Gamma, d\Gamma])$ be the interval rough number, then the expected value of $\varepsilon$ is:

$$E(\varepsilon) = \frac{a\Gamma + b\Gamma + c\Gamma + d\Gamma}{4} \quad (20)$$

Step 2: Standardize the matrix $R$. The standardized method is as follows to obtain the matrix $M$

$$\frac{a_{ij}\Gamma - \min_i\{c_{ij}\Gamma\}}{\max_i\{d_{ij}\Gamma\} - \min_i\{c_{ij}\Gamma\}} \quad (21)$$

In the formula, $a_{ij}\Gamma$ can be replaced by $b_{ij}\Gamma$, $c_{ij}\Gamma$ and $d_{ij}\Gamma$. Likewise, the four endpoints of $\varepsilon_{ij}$ are standardized in turn, and Equation (21) is used to convert the interval rough numbers into the expected values, as shown in Table 9:

Step 3: Construct the comprehensive influence matrix. Let the direct expected matrix between forest farmers' credit risk factors under internet crowdfunding be $G\Gamma$, $G\Gamma = (g_{ij}\Gamma)_{n \times n}$, and standardize the direct influence matrix $G\Gamma$ to obtain the standardized direct matrix $\hat{G}\Gamma$:

$$l\Gamma = \frac{1}{\max_{1 \le i \le n} \sum_{j=1}^{n} g_{ij}\Gamma}, \ \hat{G}\Gamma = l\Gamma G\Gamma \quad (22)$$

Then construct the comprehensive influence matrix $H_\tau$, as shown in formula (15):

$$H_\tau = \hat{G}\Gamma(E - \hat{G}\Gamma)^{-1} \quad (23)$$

Calculate the matrix by MATLAB software to obtain the comprehensive influence matrix, as shown in Table 10.

**Table 9.** Expected value matrix.

| Influencing Factors | $D_{11}$ | $D_{12}$ | $D_{21}$ | $D_{22}$ | $D_{23}$ | $D_{24}$ | $D_{25}$ | $D_{31}$ | $D_{32}$ | $D_{33}$ | $D_{34}$ |
|---|---|---|---|---|---|---|---|---|---|---|---|
| $D_{11}$ | 0.0000 | 0.0780 | 0.4267 | 0.2333 | 0.3367 | 0.0461 | 0.1443 | 0.6452 | 0.3771 | 0.0810 | 0.3533 |
| $D_{12}$ | 0.2233 | 0.0000 | 0.4608 | 0.5433 | 0.5554 | 0.3767 | 0.3271 | 0.4427 | 0.4267 | 0.4726 | 0.2386 |
| $D_{21}$ | 0.6325 | 0.5233 | 0.0000 | 0.7129 | 0.4536 | 0.3657 | 0.2356 | 0.4529 | 0.4608 | 0.3867 | 0.2871 |
| $D_{22}$ | 0.6733 | 0.7829 | 0.6186 | 0.0000 | 0.7367 | 0.7986 | 0.8671 | 0.7642 | 0.6833 | 0.7557 | 0.6471 |
| $D_{23}$ | 0.7067 | 0.5867 | 0.1486 | 0.3733 | 0.0000 | 0.4643 | 0.4633 | 0.4267 | 0.4533 | 0.2667 | 0.6643 |
| $D_{24}$ | 0.8633 | 0.8521 | 0.7443 | 0.9614 | 0.7067 | 0.0000 | 0.8671 | 0.7800 | 0.5333 | 0.8071 | 0.7680 |
| $D_{25}$ | 0.2225 | 0.2410 | 0.4267 | 0.5733 | 0.6033 | 0.4529 | 0.0000 | 0.6467 | 0.4733 | 0.3971 | 0.2671 |
| $D_{31}$ | 0.5680 | 0.2643 | 0.5408 | 0.4048 | 0.4592 | 0.6433 | 0.4000 | 0.0000 | 0.2354 | 0.3776 | 0.3467 |
| $D_{32}$ | 0.4864 | 0.5952 | 0.6871 | 0.7614 | 0.7067 | 0.6810 | 0.6833 | 0.8667 | 0.0000 | 0.6592 | 0.7386 |
| $D_{33}$ | 0.7480 | 0.6967 | 0.7636 | 0.8667 | 0.5871 | 0.7689 | 0.7986 | 0.7002 | 0.7048 | 0.0000 | 0.8769 |
| $D_{34}$ | 0.3129 | 0.2158 | 0.7067 | 0.5680 | 0.4608 | 0.4567 | 0.3771 | 0.5067 | 0.3567 | 0.2767 | 0.0000 |

**Table 10.** Comprehensive influence matrix.

| Influencing Factors | $D_{11}$ | $D_{12}$ | $D_{21}$ | $D_{22}$ | $D_{23}$ | $D_{24}$ | $D_{25}$ | $D_{31}$ | $D_{32}$ | $D_{33}$ | $D_{34}$ |
|---|---|---|---|---|---|---|---|---|---|---|---|
| $D_{11}$ | 0.0654 | 0.0662 | 0.1158 | 0.0969 | 0.1060 | 0.0680 | 0.0774 | 0.1487 | 0.0996 | 0.0633 | 0.1028 |
| $D_{12}$ | 0.1334 | 0.0945 | 0.1586 | 0.1767 | 0.1726 | 0.1438 | 0.1393 | 0.1700 | 0.1417 | 0.1435 | 0.1291 |
| $D_{21}$ | 0.1868 | 0.1615 | 0.1113 | 0.2018 | 0.1681 | 0.1477 | 0.1345 | 0.1803 | 0.1518 | 0.1390 | 0.1407 |
| $D_{22}$ | 0.2542 | 0.2467 | 0.2489 | 0.1900 | 0.2657 | 0.2560 | 0.2663 | 0.2872 | 0.2318 | 0.2334 | 0.2418 |
| $D_{23}$ | 0.1893 | 0.1626 | 0.1286 | 0.1602 | 0.1103 | 0.1542 | 0.1560 | 0.1739 | 0.1468 | 0.1212 | 0.1796 |
| $D_{24}$ | 0.2858 | 0.2631 | 0.2737 | 0.3103 | 0.2736 | 0.1735 | 0.2756 | 0.3013 | 0.2257 | 0.2472 | 0.2643 |
| $D_{25}$ | 0.1395 | 0.1289 | 0.1603 | 0.1856 | 0.1837 | 0.1581 | 0.1050 | 0.1996 | 0.1512 | 0.1396 | 0.1377 |
| $D_{31}$ | 0.1740 | 0.1258 | 0.1689 | 0.1616 | 0.1614 | 0.1715 | 0.1465 | 0.1182 | 0.1201 | 0.1318 | 0.1411 |
| $D_{32}$ | 0.2250 | 0.2180 | 0.2474 | 0.2679 | 0.2526 | 0.2357 | 0.2374 | 0.2873 | 0.1436 | 0.2153 | 0.2432 |
| $D_{33}$ | 0.2662 | 0.2403 | 0.2700 | 0.2940 | 0.2533 | 0.2564 | 0.2622 | 0.2850 | 0.2378 | 0.1493 | 0.2705 |
| $D_{34}$ | 0.1457 | 0.1219 | 0.1887 | 0.1812 | 0.1630 | 0.1532 | 0.1451 | 0.1791 | 0.1346 | 0.1221 | 0.0994 |

Step 4: Calculate the centrality degree ($zxm_i$)and the causality degree ($zxu_i$) according to the comprehensive influence matrix of forest farmers' credit risk factors under internet crowdfunding, where $ZXD_i$ is the influencing degree and is the influenced degree.

$$ZXD_i = \sum_{j=1}^{n} t_{ij}, i = 1, 2, 3, \cdots, n \tag{24}$$

$$ZXR_i = \sum_{i=1}^{n} t_{ij}, i = 1, 2, 3, \cdots, n \tag{25}$$

$$zxm_i = ZXD_i + ZXR_i (i = 1, 2, \cdots, n) \tag{26}$$

$$zxu_i = ZXD_i - ZXR_i (i = 1, 2, \cdots, n) \tag{27}$$

According to the above analysis, the centrality degree and the causality degree are calculated by Matlab using Formulas (24)–(27). The specific results are shown in Table 11.

**Table 11.** Centrality degree and causality degree of credit risk factors.

| Sequence Number | $D_{11}$ | $D_{12}$ | $D_{21}$ | $D_{22}$ | $D_{23}$ | $D_{24}$ | $D_{25}$ | $D_{31}$ | $D_{32}$ | $D_{33}$ | $D_{34}$ |
|---|---|---|---|---|---|---|---|---|---|---|---|
| Centrality Degree | 3.0754 | 3.4327 | 3.7955 | 4.9482 | 3.7931 | 4.8120 | 3.6347 | 3.9515 | 4.3582 | 4.4910 | 3.5841 |
| Causality Degree | −1.0551 | −0.2264 | −0.3487 | 0.4958 | −0.4277 | 0.9760 | −0.2561 | −0.7096 | 0.7886 | 1.0794 | −0.3162 |

According to the values of the centrality degree and the causality degree, a cause-and-effect diagram can be made as shown in Figure 1.

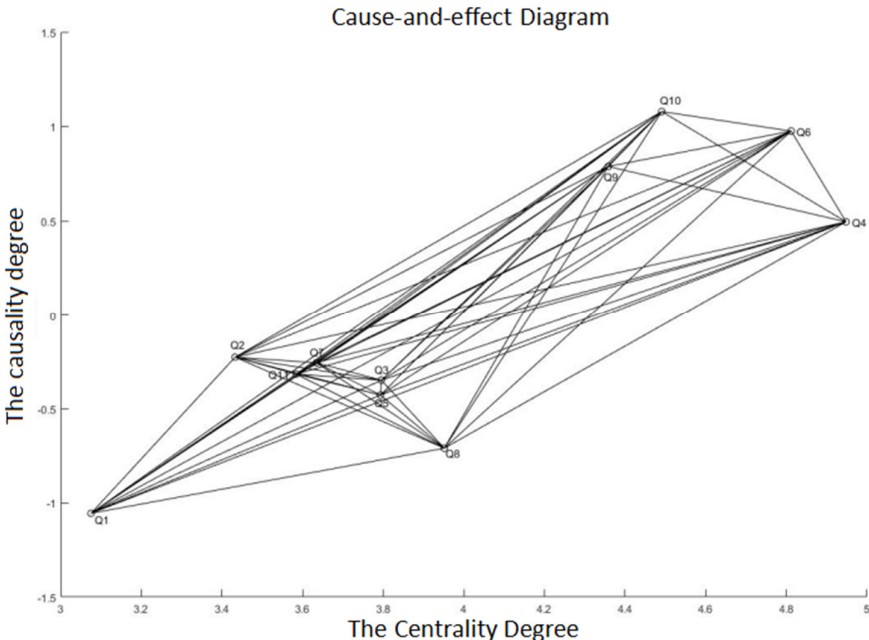

**Figure 1.** Causal relationship of the comprehensive influence of credit risk factors.

## 3. Results

Based on the results of the combined weight coefficient and interval rough number DEMATEL method, the product of the centrality value and each weight of credit risk indicators are calculated to obtain each comprehensive influence degree of forest farmers' credit risk under internet crowdfunding. The comprehensive influence degree can accurately describe the importance of forest farmers' credit risk factors and reduce the subjective one-sidedness of the combined weight coefficient and interval rough number DEMATEL method. The formula is as follows:

$$z_\hbar \Gamma = Lm_\hbar \Gamma \cdot w_\hbar \Gamma, (\hbar = 1, 2, \cdots, 11) \tag{28}$$

In which $w_\hbar \Gamma$ is the weight value of forest farmers' risk measurement indicator by the combined subjective-objective weighting coefficient method. The calculation results are shown in Table 12:

**Table 12.** Comprehensive influence degree of forest farmers' credit risk factors under internet crowdfunding.

| Influencing Factors | $D_{11}$ | $D_{12}$ | $D_{21}$ | $D_{22}$ | $D_{23}$ | $D_{24}$ | $D_{25}$ | $D_{31}$ | $D_{32}$ | $D_{33}$ | $D_{34}$ |
|---|---|---|---|---|---|---|---|---|---|---|---|
| $z_{\hbar}$ | 0.1397 | 0.1466 | 0.3395 | 0.4705 | 0.2682 | 0.9509 | 0.1620 | 0.2892 | 0.4056 | 0.7919 | 0.2576 |
| Final Weight | 0.0331 | 0.0347 | 0.0804 | 0.1114 | 0.0635 | 0.2252 | 0.0385 | 0.0685 | 0.0961 | 0.1876 | 0.0610 |
| Ranking | 11 | 9 | 5 | 3 | 7 | 1 | 10 | 6 | 4 | 2 | 8 |

Through the calculation of the above comprehensive influence degree, it is concluded that the degree of investor information asymmetry, the intensity of supervision, the degree of innovation and cooperation between funders and investors are the main credit risk factors of forest farmers under internet crowdfunding mode.

In practice, due to forest farmers' lack of mortgage assets, the lender usually provides loans according to the credit risk status of the borrower (forest farmers) and can give large loans to forest farmers with high reputations and low credit risk. This paper constructs a forest farmer credit risk measurement model based on a combination weighting approach. The practical application of the model is exemplified as follows:

In reality, three forest farmers from different regions and in different loan environments are randomly selected as QY1, QY2 and QY3. The lender provides loans according to the credit risk status of these three forest farmers.

According to the results of the credit risk measurement model, the credit score value of the credit evaluation indicator is established (the total credit score is set as 10,000 points). The credit evaluation of the three forest farmers is based on the credit risk indicator evaluation and indicator weight; according to the scoring results (based on the principle that the smaller the credit risk, the higher the credit score), the lender determines whether to lend and the amount of loan, etc. The lender invites experts in relevant fields to give scores (credit scores) according to the actual situation of the forest farmers' credit indicator evaluation system, and obtains the credit evaluation results of the three forest farmers, as shown in Table 13.

**Table 13.** Results of the credit evaluation of the three forest farmers.

| Indicator Evaluation | Corresponding Credit Score | QY1 (Credit Score) | QY2 (Credit Score) | QY3 (Credit Score) |
|---|---|---|---|---|
| Investor's cognitive ability ($D_{11}$) | 331 | 291 | 288 | 297 |
| Investor's supervision ability ($D_{12}$) | 347 | 332 | 331 | 332 |
| Financing duration and scope ($D_{21}$) | 804 | 757 | 754 | 759 |
| Degree of innovation and cooperation with the investor ($D_{22}$) | 1114 | 1024 | 1021 | 1022 |
| Degree of information asymmetry with the crowdfunding platform ($D_{23}$) | 635 | 633 | 628 | 629 |
| Degree of information asymmetry with the investor ($D_{24}$) | 2252 | 1966 | 1864 | 1958 |
| Withdrawal difficulty ($D_{25}$) | 385 | 327 | 326 | 350 |
| Margin ratio ($D_{31}$) | 685 | 537 | 540 | 588 |
| Service fee proportion ($D_{32}$) | 961 | 863 | 856 | 854 |
| Intensity of regulation ($D_{33}$) | 1876 | 193 | 190 | 185 |
| Number of media reports ($D_{34}$) | 610 | 552 | 554 | 551 |
| Total | 10000 | 7475 | 7352 | 7525 |

Assuming other external factors are not considered, the lender determines the amount of the loan according to the credit score of forest farmers' credit evaluation. To a certain extent, the higher the credit score, the larger amount the lender loans to the forest farmer; forest farmers with low credit scores will not be given loans or be given a small number of loans.

It can be seen from the above results that according to the credit evaluation results of the three forest farmers, the loan amount of the lender to forest farmer 3 is the largest. When the lender's funds are limited, priority will also be given to forest farmer 1, as the lender's evaluation credit score for forest farmer 1 reaches 7475.

## 4. Discussion and Implications

In order to control the risk of fund withdrawal, the lender needs to evaluate the credit status of forest farmers applying for loans; the construction of forest farmers' credit risk measurement model not only makes an effective quantitative evaluation of forest farmers, but provides a practical basis for the connection between the measurement model and the real situation, but also provides credit strategies for lenders.

The immunity system of forest farmers mainly includes five levels: protection layer, detection layer, response layer, intrusion tolerant layer and recovery layer. The protection layer is a management system of enterprises, which is mainly used to prevent the entry of dissidents; the detection layer is a risk warning system of enterprises, which is mainly used to identify internal and external dissidents; the response layer is an emergency management system; intrusion tolerant layer is mainly used to maintain the orderly operation of core businesses; the main function of the recovery layer is self-healing ability [45,46]. With blockchain 3.0 technology, the core of the internet of value, each information or byte representing a value on the internet can be stored and measured, and its property rights can be confirmed, so as to realize the traceability of assets on the blockchain, effectively control the credit risk and improve the credit risk immunity level of forest farmers. According to the above evaluation results of forest farmers' credit risk, the main ways to enhance forest farmers' immunity to credit risk under internet crowdfunding modes are to improve adaptability through specific immunity and to improve stability through non-specific immunity.

- Promotion countermeasures at the level of specific immunity:

The government should optimize the support mode of special funds for blockchain projects, implement various preferential policies for forest farmers, and give forest farmers room for trial and error. Combining the consensus on blockchain 3.0 technology with an incentive mechanism, the government should prompt the formulation of consensus rules concerning responsibilities, rights and benefits, forming an effective combination of market incentive, policy incentive and resources incentive, thus effectively reducing forest farmers' credit risk and continuously generating innovation power to improve the defense ability and the immunity level of forest farmers to credit risk.

- Promotion countermeasures at the level of nonspecific immunity:

The application of blockchain 3.0 technology relies on a reliable internet connection environment. However, in some remote areas of China, forest farmers do not have access to the infrastructure of blockchain, let alone using blockchain 3.0 technology. Therefore, the primary task is to improve the most basic conditions for technology application. Memory mechanism is positively correlated with learning ability to a certain extent. The blockchain 3.0 technology with its strong memory ability can effectively eliminate the credit risk of forest farmers. Therefore, the government and enterprises should work together [47,48] to build up throughout the internet the essential infrastructure for blockchain technology, to utilize blockchain 3.0 technology to create a learning organization for forest farmers and strengthen their memory mechanism [49,50]. The primary task of strengthening the memory mechanism of forest farmers is to build a healthy growth mech-

anism under the agglomeration of enterprises [51–56]. The specific growth mechanism is shown in Figure 2:

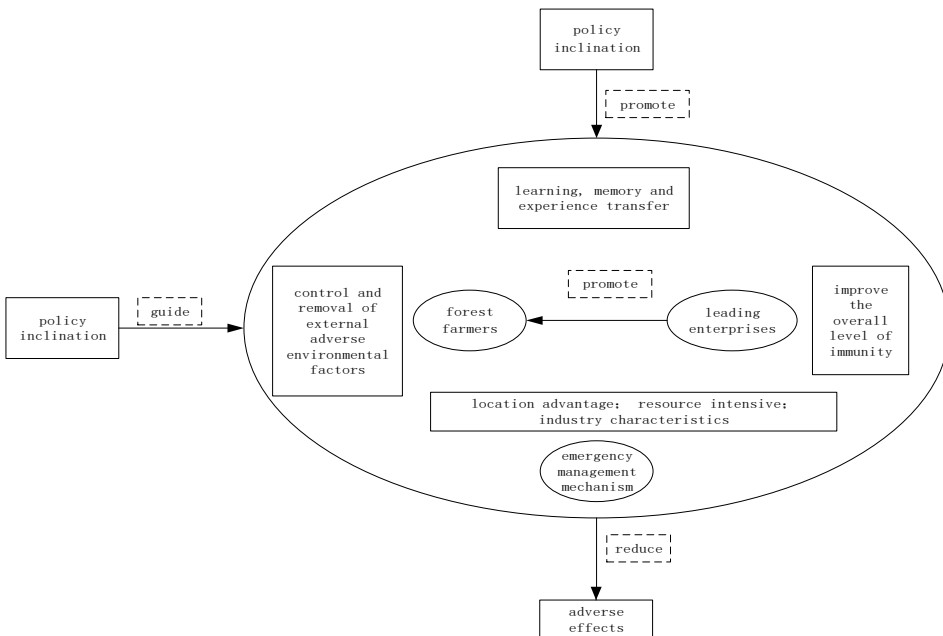

**Figure 2.** Forest farmers' growth mechanism based on theory of organizational immunity under enterprise agglomeration.

Enhancing the ability of organizational learning can facilitate the immune system of forest farmers to produce immune antibodies to resist various credit risk factors and constructing a healthy growth mechanism on this basis can fundamentally reduce the credit risk and improve the immunity of forest farmers under internet crowdfunding modes [57,58]. In addition, in view of the sudden impact on the external environment, the government should build an emergency management mechanism for online crowdfunding platforms to reduce the adverse impact of emergencies on forest farmers.

## 5. Conclusions

Online crowdfunding has become a research hotspot of scholars in relevant fields with its rapid development in recent years, but now there are problems, such as a long withdrawal cycle and high communication cost. At present, the problems with the credit risk of small and micro forest farmers are mainly as follows: it is difficult to objectively obtain the credit risk evaluation indicator system of forest farmers' network crowdfunding; there are few credit risk measurement methods applicable to the network crowdfunding model.

This study integrates quantitative and qualitative methods and constructs the forest farmers' credit risk measurement model using the improved DEMATEL method and combined weighting method. On the basis of previous research, by integrating the new viewpoint of the ecological health of medical immunity and drawing on organizational immunity theory, risk management theory and blockchain thinking, etc., this study constructs the growth mechanism of forest farmers based on organizational immunity theory under enterprise agglomeration.

Based on the combination weighting method, this paper calculates the weight of various credit risk factors of forest farmers and constructs a feasible credit risk evaluation model. Using precise mathematical logic, this measurement model further emphasizes and accurately describes the objective attributes, such as the fuzziness of credit risk evaluation indicators and the difficulty of digital expression. By effectively combining qualitative analysis with quantitative research, this study makes the process of the credit risk evaluation more practical, and this combined weight measurement method, effectively solves

problems, such as the difficulty in accurately measuring the credit risk under internet crowdfunding mode, integrates the decision-making thinking of different experts to avoid the subjectivity of evaluation, and thus reveals the most critical factors affecting the credit risk. In this study, the establishment of forest farmers' credit risk evaluation and control mechanism under the network crowdfunding mode effectively improves the risk control level of forest farmers and enhances their ability to prevent and resolve credit risks.

However, there are some limitations in the data mining of forest farmers' credit risk in this research. Due to the lack of massive data and the difficulty in starting and obtaining some data in the research process, this study fails to deeply mine the credit risk data with machine learning, deep learning or some other methods. Because of the continuous updating of statistical data of relevant departments and the difficulty of obtaining forest farmers' data under the network crowdfunding mode, some research in this study can only be conducted based on the analysis of the statistical data available.

The direction and prospect of further research are as follows: exploring the credit risk threshold that is easy to identify, and taping the threshold of forest farmers' ability to resist credit risk; identifying the defaulting forest farmers and non defaulting forest farmers to the greatest extent using the optimization theory and method; making an in-depth and systematic study on the theory of forest farmers' credit risk management by the combined use of the modern management theory, system engineering theory, etc.

Further research can be conducted on credit risk control based on machine learning, big data technology and blockchain technology. By the use of big data risk control technology and artificial intelligence to control the credit risk of forest farmers to the greatest extent, the stable development of forestry financial technology can be promoted. In the future, studies can focus on credit risk measurement, pricing and management practice of forest farmers' network crowdfunding under blockchain technology, as well as credit risk default probability and rating transfer model based on machine learning and deep learning.

**Author Contributions:** All of the authors contributed to conceptualization, formal analysis, investigation, methodology, and writing and editing of the original draft. All authors have read and agreed to the published version of the manuscript.

**Funding:** This research was funded by the Zhejiang Provincial Planning Projects of Philosophy and Social Science (21NDQN300YB) and the Social Science Foundation of National Radio and Television Administration, PRC (GDT2004).

**Institutional Review Board Statement:** Not applicable.

**Informed Consent Statement:** Informed consent was obtained from the respondents of the survey.

**Data Availability Statement:** The data will be made available on request from the corresponding author.

**Conflicts of Interest:** The authors declare no conflict of interest.

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
