# Peer review of "Credit Risk Evaluation of Forest Farmers under Internet Crowdfunding Mode: The Case of China’s Collective Forest Regions"

_sustainability, doi:10.3390/su14105832_

Round 1

Reviewer 1 Report

I'm fine with the current version.

Author Response

Dear Sir or Madam:

Thank you very much for the nice comments and your continuous support. I will carefully check the English spell and expression again. I and my team will continue to do the further research in this field.

Thank you again.

Best regards

Your sincerely

Yanxiong, Wu

Reviewer 2 Report

I appreciate the changes made to the article proposal following the previous evaluation made and as a consequence I recommend its publication.

Author Response

Dear Sir or Madam:

Thank you very much for the nice comments and your continuous support. I will carefully check the English spell and expression again. I and my team will continue to do the further research in this field.

Thank you again.

Best regards

Your sincerely

Yanxiong, Wu

This manuscript is a resubmission of an earlier submission. The following is a list of the peer review reports and author responses from that submission.

Round 1

Reviewer 1 Report

Please see the attached report.

Reviewer 2 Report

I appreciate the quantitative model developed in this article proposal.

However, I recommend the extension of the bibliography, which seems a bit small compared to the research dimension that the topic developed in this article proposal.

Reviewer 3 Report

My main remark is about the lack of care in writing formulas and formatting the manuscript. The problem is, among others the lack of clarity of the mathematical records in section 2. All mathematical records, as well as the accompanying text descriptions, are sloppy and therefore illegible (e.g. mathematical symbols shifted upwards in relation to the text line). Also, Figure 1 is illegible. In addition, the authors introduce abbreviations without explaining them (e.g., LFPP, AMF, EDM, and even AHP).

All problems related to the illegibility of the article and the lack of diligence in preparing the manuscript must be corrected. An article cannot be accepted if the authors do not take the reader seriously and fail to correctly format the mathematical notations.

In addition:

- The authors introduce "improved analytic hierarchy process" without explaining what the improvements are in relation to the classic AHP method.

- The introduction should contain research questions and the research goal, as well as discuss the structure of the article.

- Conclusions should indicate the directions of further research and research limitations.